# Flavonoids with Glutathione Antioxidant Synergy: Influence of Free Radicals Inflow

**DOI:** 10.3390/antiox9080695

**Published:** 2020-08-03

**Authors:** Igor Ilyasov, Vladimir Beloborodov, Daniil Antonov, Anna Dubrovskaya, Roman Terekhov, Anastasiya Zhevlakova, Asiya Saydasheva, Vladimir Evteev, Irina Selivanova

**Affiliations:** 1Department of Chemistry, Sechenov First Moscow State Medical University, Trubetskaya Str. 8/2, 119991 Moscow, Russia; Beloborodov_v_l@staff.sechenov.ru (V.B.); Antonov_d_o@student.sechenov.ru (D.A.); anna.mma@list.ru (A.D.); terekhov_r_p@staff.sechenov.ru (R.T.); Zhevlakova_a_k@staff.sechenov.ru (A.Z.); Saydasheva_a_n@student.sechenov.ru (A.S.); Selivanova_i_a@staff.sechenov.ru (I.S.); 2Federal State Budgetary Institution “Scientific Centre for Expert Evaluation of Medicinal Products” of the Ministry of Health of the Russian Federation, Petrovsky blvd. 8/2, 127051 Moscow, Russia; Evteev@expmed.ru

**Keywords:** glutathione, taxifolin, quercetin, rutin, morin, ABTS, antioxidant activity, synergistic effect

## Abstract

This report explores the antioxidant interaction of combinations of flavonoid–glutathione with different ratios. Two different 2,2′-azino-bis(3-ethylbenzothiazoline-6-sulfonic acid radical (ABTS^•+^)-based approaches were applied for the elucidation of the antioxidant capacity of the combinations. Despite using the same radical, the two approaches employ different free radical inflow systems: An instant, great excess of radicals in the end-point decolorization assay, and a steady inflow of radicals in the lag-time assay. As expected, the flavonoid–glutathione pairs showed contrasting results in these two approaches. All the examined combinations showed additive or light subadditive antioxidant capacity effects in the decolorization assay. This effect showed slight dilution dependence and did not change when the initial ABTS^•+^ concentration was two times as high or low. However, in the lag-time assay, different types of interaction were detected, from subadditivity to considerable synergy. Taxifolin–glutathione combinations demonstrated the greatest synergy, at up to 112%; quercetin and rutin, in combination with glutathione, revealed moderate synergy in the 30–70% range; while morin–glutathione appeared to be additive or subadditive. In general, this study demonstrated that, on the one hand, the effect of flavonoid–glutathione combinations depends both on the flavonoid structure and molar ratio; on the other hand, the manifestation of the synergy of the combination strongly depends on the mode of inflow of the free radicals.

## 1. Introduction

As a theoretical consideration, a combination of two or more antioxidants might form a “complex network” with cooperative component interactions [1,2,3]. Thus, deviations of the experimentally derived antioxidant capacities from the theoretically calculated values can be observed, giving rise to the synergistic or subadditive (also known as antagonistic or negative synergistic) effects. Apparently, this phenomenon can substantially influence foods, nutrients, and pharmaceuticals as well as biological systems. Despite the potential sophistication of the antioxidant “complex network” systems, the elementary model of the interaction is a binary combination. 

As well as flavonoids, which are common components of fruits, vegetables, tea, herbs, and wine and reveal a wide variety of biological activities [4,5,6,7], glutathione can be supplied by foods [8] or supplements, thus improving glutathione status [9,10,11,12]. Glutathione (γ-l-glutamyl-l-cysteinyl-glycine, GSH)—a tripeptide found in most living cells from bacteria to mammals [13]—is the classical scavenging antioxidant which is essential for protecting hepatocytes, erythrocytes, and other cells against toxic injury via enzymatic and nonenzymatic reactions; it acts with electrophilic xenobiotics as a low-molecular-weight detoxifier, transition metals chelator, redox-mediator of proteins’ posttranslational modification by glutathionylation, etc. [14,15,16,17,18]. The relatively high glutathione concentration (millimolar range) together with the high ratio of glutathione/glutathione disulfide (GSH/GSSG) in healthy cells (more than 98% of the reduced form) makes the GSSG/GSH redox couple the principal cellular redox buffer and the indicator of the cellular redox state/antioxidant defense against oxidative/nitrosative stress and radiation damage [18,19,20,21]. An important factor to consider is the possible flavonoid–glutathione interaction in food/beverages or taken as food supplements and in therapeutic products. However, the effect of their co-presence on each other’s biological activity, including antioxidant activity (AOA), is not yet completely understood.

Here, we focused on the elucidation of the antioxidant capacity of flavonoid–glutathione binary combinations. We employed different flavonoid/GSH ratios—from equal to a great excess of GSH. Although the total dietary flavonoid intake can be greater than GSH, the expected concentration of individual flavonoids in an organism is always much less than GSH [8,22,23,24,25,26,27,28,29,30,31]. 

Quercetin, rutin (a glycoside of quercetin), morin (a structural isomer of quercetin), and taxifolin (a reduced form of quercetin) were used as flavonoid components for the combinations (see Figure 1). These flavonoids are well-known natural antioxidants whose potential health benefits have attracted a great deal of interest in their studies as prospective drugs [7,32,33,34,35,36,37,38,39]. Thus, a large and growing body of literature has been devoted to the modification of their biopharmaceutical and physicochemical properties, such as antioxidant activity and bioavailability, by crystal engineering [40,41,42,43,44], obtaining solid dispersions [45,46,47,48,49], and the development of new polymorphic forms [50,51,52,53,54]. This report is in line with the abovementioned works and aims to study the possible alteration of the antioxidant activity of flavonoids by combining them with other natural antioxidants. Instead of applying completely different antioxidant assays, which often result in unrelated and even controversial data that are difficult to interpret, we focused on two different 2,2′-azino-bis[3-ethylbenzothiazoline-6-sulfonic acid radical (ABTS^•+^) radical–cation scavenging approaches based on the decolorization end-point or lag-time experimental strategies [55,56].

## 2. Materials and Methods 

### 2.1. Materials

#### 2.1.1. Components of Flavonoid–Glutathione Combinations 

Taxifolin (LLC “Flavir”, Irkutsk, Russia), reduced glutathione (GSH) and morin were purchased from Sigma-Aldrich Chemie Gmbh, Steinheim, Germany. Rutin was purchased from Merck, Darmstadt, Germany; quercetin was purchased from Asha Organics, Geel, Belgium.

#### 2.1.2. Reagents 

ABTS, 2,2’-azinobis(3-ethylbenzothiazoline-6-sulfonic acid) diammonium salt, potassium persulfate (di-potassium peroxydisulfate, PP), *N*,*N*-dimethylformamide and the components of phosphate buffer saline pH 7.0 (PBS)—i.e., dipotassium hydrogen phosphate, potassium dihydrogen phosphate and sodium chloride—were obtained from Sigma-Aldrich. High performance liquid chromatography (HPLC)-grade ethanol and *N*,*N*-dimethylformamide (puriss.) were obtained from Merck Ltd., Germany and JSC LenReactiv, Russia, respectively.

All reagents were prepared on the day of use. Stock solutions of taxifolin, quercetin, and morin were prepared in ethanol; rutin was initially dissolved in a small amount of *N*,*N*-dimethylformamide and then ethanol (solvent ratio 1:10) and GSH in deionized water. None of the solvents interfered with the assays. 

### 2.2. Methods 

Spectrophotometric measurements were performed on a Cary 100 spectrophotometer (Varian, Palo Alto, CA, USA) with a 1 cm Peltier temperature-controlled cuvette. 

#### 2.2.1. Decolorization Assay

**General procedure.** The decolorization assay was performed as reported by Re et al. [56] with minor modifications. Briefly, the antioxidant solution was added to the pre-generated ABTS^•+^, shaken intensively for 15 s and placed in a heating chamber at 30 °C for up to 30 min in the dark. The addition of antioxidants led to the characteristic ABTS^•+^ maxima of absorption decreasing, which were read at 730 nm at the first, 10th, 15th, 20th, 25th, and 30th minute. The appropriate initial ABTS^•+^ absorbance without antioxidants was adjusted to 1.00 ± 0.05 for all measurements and read for each sample just before the addition of antioxidants. The quercetin–glutathione 1:5 combination was additionally tested at 0.05, 0.75, and 2.0 ABTS^•+^ initial absorbance to elucidate the influence of ABTS^•+^ concentration on the observed antioxidant capacity. 

**Generation of ABTS^•+^.** To generate ABTS^•+^, the water stock solutions of ABTS and PP were mixed to final concentrations of 7 mM and 2.5 mM, respectively, and allowed to stand at room temperature overnight. The stock ABTS^•+^ was then used for 1–2 days (kept at a storage temperature of 5 °C). 

**ABTS^•+^ self-bleaching.** We examined the radical–cation self-bleaching of ABTS^•+^ at concentrations giving different initial absorbances—from 0.5 to 2.0—at 730 nm. After dilution of a corresponding volume of stock ABTS^•+^ in PBS, the initial absorbance and then at 1, 10, 15, 20, 25, and 30 min was measured.

***n*-value and mixture effect calculations.** The decrease in the ABTS^•+^ absorbance ∆A and the inhibition percentage of ABTS^•+^, Inh%, were calculated applying the following equations:∆A = A_0_ − A_1_ − ∆A_self-bleaching_(1)
Inh% = 100% × ∆A/A_0_,(2)
where A_0_ and A_1_ are the ABTS^•+^ absorbances at 730 nm, a pathlength 1 cm at a zero time-point and after a certain time of incubation with antioxidant (antioxidant mixture), respectively. The self-bleaching of ABTS^•+^ ∆A_self-bleaching_ was calculated as the difference between the ABTS^•+^ absorbance at a zero time-point and after a certain time of incubation without antioxidant.

The amount of ABTS^•+^ radicals scavenged by one molecule of antioxidant (*n*-value) in the decolorization assay was calculated as follows:*n*-value = C_ABTS_^•+^/C_antioxidant_ = ∆A/ε × C_antioxidant_ × l,(3)
where C_ABTS_^•+^ is the concentration of the scavenged ABTS^•+^, ε is the extinction coefficient at 15,000 L*mol^−1^*cm^−1^ (at 730 nm), l is the optical pathlength of 1 cm and C_antioxidant_ is the concentration of antioxidant.

The mixture effect, *ME*, was calculated in two ways:
1.The “traditionally calculated mixture effect”:ME_traditional_ = (∆A^mix^_experimental_ − ∆A^mix^_theoretical_) × 100%/∆A^mix^_theoretical_,(4)
where ∆A^mix^_experimental_ and ∆A^mix^_theoretical_ were calculated as follows: ∆A^mix^_experimental_ = A^mix^_0_ − A^mix^_30_ + ∆A_self-bleaching_(5)
∆A^mix^_theoretical_ = A^mix^_0_ − (A^1^_30_+A^2^_30_ − A_0_) + ∆A_self-bleaching_(6)
where A^1^_30_ and A^2^_30_ is the absorbance at the 30th min of component 1 and component 2, respectively, and A^mix^_0_ is the initial absorbance of the mixture of these components.2.The “Webb’s simulation mixture effect” [57]:ME_Webb’s simulation_ = (Inh%^mix^_experimantal_ − Inh%_Webb’s simulation_)*100%/Inh%_Webb’s simulation_,(7)
where Inh%^mix^_experimental_ and Inh%_Webb’s simulation_ were calculated as follows:Inh%^mix^_experimental_ =100%•∆A^mix^_experimental_/A^mix^_0_(8)
Inh%_Webb’s simulation_ = [1 − (1 − Inh%_flavonoid_/100) × (1 − Inh%_glutathione_/100)] × 100%.(9)

#### 2.2.2. Lag-Time Assay

**General procedure.** The lag-time assay was performed as reported by Ilyasov et al. [55]. Briefly, a solution of PP was added to the antioxidant and ABTS solution in PBS, and then the ABTS^•+^ absorbance was continuously read until ABTS^•+^ accumulation recommenced and became steady.

The kinetics of the ABTS^•+^ accumulation were monitored at maxima of 730 nm with an incubation temperature of 22 ± 1 °C. For each measurement, 4–50 µL of antioxidant stock solution was added to 3.3 mL of ABTS dissolved in PBS, and then the reaction was started by the addition of 0.7 mL of PP dissolved in PBS with subsequent intensive shaking for 15 s; final concentrations of ABTS and PP were 1.21 mM and 0.43 mM. In the absence of antioxidants, the accumulation of ABTS^•+^ began as soon as the PP was added. Normally, absorbance reached 1.25 ± 0.05 by the 15th min of incubation and continually increased for several hours. When an antioxidant was added, the lag-time was usually observed before the ABTS^•+^ accumulation commenced. The antioxidant concentrations were adjusted to induce a lag-time of not less than 3 min in order to decrease the influence of preparatory time error. 

***n*-value and mixture effect calculations.** The amount of ABTS^•+^ radicals scavenged by one molecule of antioxidant (*n*-value) for the lag-time assay results was calculated as follows: *n*-value = C_expected ABTS_^•+^/C_antioxidant_,(10)
where C_expected ABTS_^•+^ is the expected accumulated concentration of ABTS^•+^, which corresponds to a certain lag-time, and C_antioxidant_ is the concentration of antioxidant.

The expected accumulated ABTS^•+^ concentration was calculated from an averaged kinetic curve, which was obtained from seven independent runs (see Figure 2a), considering an ABTS^•+^ radical–cation λmax of 730 nm and ε at 15,000 L*mol^−1^*cm^−^^1^. 

The mixture effect was calculated in two ways:
The first one was the traditional method, which took into account the lag-time duration, giving the “lag-time mixture effect”:ME_lag-time_ = [(lag-time_experimental_ − lag-time_theoretical_)/lag-time_theoretical_] × 100%,(11)
where lag-time_experimental_ is the experimentally derived lag-time duration of the examined combination, in min; lag-time_theoretical_ is the calculated sum of lag-times of individual components of the examined combination, in min.The second was calculated through the absolute ABTS^•+^ concentration scavenged by antioxidants compared with the expected accumulated ABTS^•+^ amount, giving the “scavenged ABTS^•+^ mixture effect”.
ME_scavenged ABTS_^•+^ = [(C_expected ABTS_^•+^_experimental_ − C_expected ABTS_^•+^_theoretical_)/C_expectedABTS_^•+^_theoretical_] × %,(12)
where C_expected ABTS_^•+^
_experimental_ and C_expected ABTS_^•+^_theoretical_ were calculated from the corresponding lag-times applying the averaged kinetic curve (Figure 2a) and taking ε as 15,000 L*mol^−1^*cm^−1^.

### 2.3. Statistics

The results are shown as the mean (± standard deviation) of at least three determinations. The paired Student’s *t*-test was used for the analysis of experimental and simulation results. The R-squared statistics and the F-test of overall significance were used for regression analysis.

## 3. Results

### 3.1. Antioxidant Capacity in the Decolorization Assay

The original protocol of the decolorization assay, which is based on the discoloration of pre-generated ABTS^•+^ radical–cations by antioxidants [56], was modified in this work: The initial ABTS^•+^ absorbance was increased from 0.7 to 1.0, and the incubation time was extended from 4–6 min to 30 min. Thus, an additional methodology adjustment was performed for clarity regarding the contribution of ABTS^•+^ self-bleaching to the measured antioxidant capacity, the influence of different initial ABTS^•+^ absorbances on the antioxidant effect of individual compounds and combinations and the susceptibility to different orders of the addition of components of a mixture effect. After that, the individual compounds and flavonoid–glutathione combinations were examined.

#### 3.1.1. Methodology Adjustment 

**ABTS^•+^ self-bleaching.** The ABTS^•+^ radical–cation absorbance decrease-time curves demonstrated gradual self-bleaching, which depended very little on the initial concentration of ABTS^•+^ and was almost equal by the 15th minute and thereafter in all experiments (see Appendix A). The relative rate of self-bleaching decreased to being almost negligible by the 30th minute in all cases, but the absolute cumulative absorbance loss (ΔAbs) was quite noticeable; i.e., 0.056, 0.061, 0.075, and 0.079 at an initial absorbance of 0.5, 0.75, 1.0 and 2.0 at 730 nm, respectively. This self-bleaching contribution was adjusted to all further antioxidant-mediated absorbance decrease measurement results.

**Antioxidant capacity measurement at different initial ABTS^•+^ absorbance.** To compare the relative antioxidant capacity of antioxidants, we calculated the number of ABTS^•+^ radicals scavenged by one molecule of antioxidant (*n*-value). At the initial ABTS^•+^ absorbance of 1.0 absorbance units, this parameter showed no dependence on the concentration of antioxidants by the 30th minute, except for morin and glutathione, for which the *n*-value dependence on AO concentration was statistically significant, although the absolute *n*-value shift was insufficient. 

The *n*-values for quercetin at different initial ABTS^•+^ absorbances—i.e., 2.0, 1.0, 0.75, and 0.5—also showed no statistically significant difference.

**Susceptibility to the different ABTS^•+^ concentrations of mixture effect.** To evaluate the susceptibility to the different ABTS^•+^ concentrations of mixture effect, we tested the same combination (quercetin–glutathione, 1:5) at various absolute and relative concentrations:(a)At different reagent concentrations, keeping their ratio the same: quercetin, glutathione and ABTS^•+^ at 0.65, 3.26 and 33.3 µM, or twice as high—i.e., 1.34, 6.52 and 66.7 µM;(b)At the same quercetin and glutathione concentrations—i.e., 1.34 and 6.52 µM, respectively—but changing ABTS^•+^ concentration: 133.2, 66.7, and 50.0 µM (with an initial absorbance of 2.0, 1.0, and 0.75, respectively).

All these combinations showed a slight subadditive effect, except for the mixture effect at the initial ABTS^•+^ absorbance of 2.0, which appeared to be additive (see Appendix A and Appendix A). **Susceptibility to the different orders of component addition of the mixture effect.** To elucidate the susceptibility to the sequential addition of components of the mixture effect, we tested the combination of quercetin–glutathione at 1:5. The concentrations of components in the incubation mixture were the same, namely 1.34, 6.52, and 66.7 µM for quercetin, glutathione, and ABTS^•+^, respectively. The introduction of components was carried out in two ways:(a)Quercetin was mixed with ABTS^•+^ and then glutathione was added 1 min later, or vice versa;(b)Glutathione was mixed with ABTS^•+^ and quercetin was added 1 min later.

In both cases, the mixture effect was subadditive and did not differ from the effect of the combination when both components were added at once (see Appendix A and Appendix A).

#### 3.1.2. Individual Compounds

All the tested flavonoids and GSH scavenged ABTS^•+^ in a two-phase manner. About 60–70% of the total ABTS^•+^ was scavenged during the first-minute fast phase. Then, the inhibition of ABTS^•+^ slowed down, leading to the second moderate phase. Nevertheless, on average, 93% and 96% of ABTS^•+^ had already been scavenged by the 10th and 15th minute, respectively. By the 30th min, the change in total ΔAbs became negligible (see Table 1 and Appendix A). The regression curve slopes rose significantly from 1 to 10 min of incubation; however, they became almost equal by the 20th–30th minute of incubation. The regression curve intercepts were rather low and even insignificant in the case of rutin and taxifolin. The antioxidant capacity of the tested compounds increased in the order glutathione < taxifolin < rutin < morin < quercetin, with *n*-values in the range of 3.5–12.4 (Table 1).

#### 3.1.3. Flavonoid–Glutathione Combinations at Different Molar Ratios

As with the previously mentioned quercetin–glutathione 1:5, the assessment of rutin–glutathione, taxifolin–glutathione, morin–glutathione, and quercetin–glutathione combinations at different ratios did not reveal considerable differences in terms of mixture effects, demonstrating subadditive interaction from the statistically insignificant value of −3% for quercetin–glutathione 1:1 to the maximal −14.5% for morin–glutathione 1:10. Webb’s simulation mixture effects were also calculated for comparison and discussion (Table 2, Appendix A).

### 3.2. Antioxidant Capacity in the Lag-Time Assay

In contrast to the decolorization assay, this approach is based on the gradual inflow and accumulation of ABTS^•+^ radical–cations as a result of the In Situ reaction of ABTS with potassium persulfate. Typically, the introduction of an antioxidant leads to the delay of ABTS^•+^ accumulation, resulting in a lag-time whose duration depends on the antioxidant concentration [55]. 

#### 3.2.1. Individual Compounds

The kinetic curve shapes for all the examined individual compounds were in line with those previously reported for this experimental design [55]. The lag-time durations for individual compounds and combinations were obtained from kinetic curves to calculate *n*-values and mixture effects, respectively (Figure 2). The calculated *n*-values were as follows: 4.7, 3.3, 2.8, 2.7, and 0.8 for quercetin, morin, rutin, taxifolin, and glutathione, respectively [58]. Combinations of the tested flavonoids with glutathione were shown to have different mixture effects, from subadditive to a considerable synergy, which depended on both the flavonoid component structure and the ratio of components. 

#### 3.2.2. Flavonoid–Glutathione Combinations at Different Molar Ratios

**Taxifolin–glutathione.** All the tested taxifolin and glutathione combinations, except for the 1:1 ratio, demonstrated significant synergy. The most considerable effect was observed at ratios from 1:8 to 1:16, with a maximum of 112% at 1:16. This synergy then gradually reduced when the glutathione content was increased but still remained significant at 1:50, amounting to about 70% (see Table 3, Appendix A).

**Quercetin–glutathione.** Quercetin with glutathione also demonstrated synergy. Again, the observed synergy rose with the glutathione content, increasing up to the 1:16 ratio and then declining. The general magnitude of synergy was less considerable than in the case of taxifolin–glutathione combinations with a maximum value which was around half the size, at 51% (see Table 3, Appendix A).

**Rutin–glutathione.** The combinations of rutin with glutathione revealed a significant synergy which generally did not depend on the component ratio. The magnitude of the mixture effect at all the examined ratios was almost the same and averaged 68 ± 11% (see Table 3, Appendix A).

**Morin–glutathione.** The only combination which showed no synergy was morin–glutathione. As in the case of the rutin–glutathione combinations, no dependence of the mixture effect on the component ratio was revealed. Additive and slight subadditive mixture effects were observed at different component ratios (see Table 3, Appendix A).

## 4. Discussion

Despite the tendency in recent years to assess the antioxidant capacity utilizing completely different methods in parallel, there are certain problems in interpreting the obtained results. In this work, we tried to apply different approaches while utilizing the same antioxidants and—more importantly—the same model radicals. There are numerous methods to measure antioxidant activity/antioxidant capacity (AOA/AOC) [3,59,60,61,62,63,64,65,66,67]; of those, the ABTS-based methods are among the most widely used [58]. We applied two approaches based on the scavenging of ABTS^•+^ radical–cations to evaluate the antioxidant interaction of flavonoid–glutathione combinations. However, several methodological considerations needed to be taken into account.

### 4.1. Methodological Considerations 

#### 4.1.1. Decolorization Assay 

Approaches based on the ABTS^•+^ discoloration have certain shortcomings which have been discussed multiple times to date [68,69,70]. One of them is ascribed to the different interlaboratory values of antioxidant capacities obtained for the same antioxidants. This can be related to complicated kinetic patterns, elevated stoichiometries and the dependence of the relative antioxidant capacity (e.g., the Trolox equivalent antioxidant capacity (TEAC) index) on antioxidant concentration [55,71,72,73]. As a result of unfinished ABTS^•+^ inhibition in the presence of antioxidants by the 4th, 6th, or even 30th minute [74,75,76,77,78,79,80], the ambiguity of the obtained data complicates their interpretation.

This impelled us to explore the optimal end-point—i.e., the incubation time when the radical inhibition has almost completed—before we proceeded to the elucidation of mixture effects. Our experiments showed that a 15–20 min incubation time turned out to be sufficient for all the tested AOs; by this time, all of them showed 96–99% of the total antioxidant capacity, even though the decrease in ABTS^•+^ still continued after 30 min of incubation at rates which were negligibly higher than self-bleaching. Since we focused on the elucidation of antioxidant capacity, we decided to adjust the starting ABTS^•+^ absorbance to 1.0 ± 0.5 instead of 0.7 ± 0.5 as used the original procedure [56]. Thus, we could use a wider range of antioxidant concentrations, as recommended by Tian and Schaich [72]. This was important for the combinations and allowed us to avoid making the lowest component concentration too low—i.e., difficult to measure in terms of the decrease of ABTS^•+^—and at the same time allowed us to work in the optimal spectrophotometer absorbance range. 

Additionally, we determined ABTS^•+^ self-bleaching parameters to account for their contribution to all measured antioxidant capacity results. In contrast to our expectations, they turned out to be slightly different at various initial ABTS^•+^ absorbances, resulting in ABTS^•+^ absorbance loss due to self-bleaching in the range from 0.056 ± 0.0014 to 0.079 ± 0.0025 by the 30th minute. However, the ABTS^•+^ absorbance decrease rates were very close at different initial ABTS^•+^ concentrations—and by the 15th minute became indistinguishable—which shows that the initial difference could be due to dissolved oxygen or traces of metals which after reacting with ABTS^•+^ ceased to influence its self-bleaching. 

The number of ABTS^•+^ radicals scavenged by one molecule of antioxidant, the *n*-value, turned out to be an extremely useful parameter to estimate the relative antioxidant capacity as it showed the stoichiometry result of the interaction of ABTS^•+^ with antioxidants. In our opinion, the lack of dependence—or slight dependence—of the same antioxidant *n*-values on concentrations in most of our experiments might be attributed not to the specific nature of the AOs but to the narrow ABTS^•+^ inhibition concentration ranges tested (10–40 inh%). In contrast, morin and glutathione were both tested in a wider range (10–60 inh%) and as a result showed a statistically significant inversely proportional decrease in *n*-value. Apparently, the higher the ABTS^•+^:antioxidant ratio, the deeper the interaction; previously, it has been shown that the reaction of antioxidants with ABTS^•+^ can be deeper when the latter is in excess [58]. However, this effect was mostly negligible. The intercepts of regression curves were small and mostly did not differ significantly from zero, which means that we reached the completion of the reaction at all the tested concentrations and therefore avoided the concentration-dependence of relative antioxidant capacity (Appendix A).

The parameter conventionally used for ABTS-based assays, the ABTS^•+^ inhibition percentage (%inh), was shown to be less reliable than the absolute absorbance loss (ΔAbs) or the *n*-value. For instance, the ΔAbs did not differ at various initial ABTS^•+^ absorbances (in the range of 0.5–2.0 absorbance units) for the same quercetin concentration, which was quite logical: A certain amount of quercetin can scavenge up to a certain amount of ABTS^•+^, independent of the total ABTS^•+^ concentration. Indeed, when the ABTS^•+^ solutions with absorbances of 1.0 or 2.0 were treated with the same amount of quercetin (with a final concentration of 1.34 µM), the absolute absorbance loss did not differ, at 0.25 ± 0.02 and 0.27 ± 0.02 absorbance units, respectively, but the calculated inhibition percentages appeared to be 24.6 ± 1.35% and 13.2 ± 0.63%, respectively. The latter result can be confusing and even misleading, meaning that the %inh parameter only works correctly if one keeps the initial absorbance at one point, allowing only minimum deviations. Thus, we based our calculations on the ΔAbs values and calculated *n*-values.

#### 4.1.2. Lag-Time Assay 

The time interval between the start of incubation and the accumulation of ABTS^•+^—the lag-time parameter—which is usually used in this type of assays was shown to be contradictory in terms of reliability. The reason for this was that the lags in our experiments appeared to be rather long and the rate of ABTS^•+^ accumulation and its change therefore began to matter. Normally, the lag time for different experiment durations can be comparable if the same events happen at the same periods during the whole period of lag. In our case, this could be only possible if the rate of radical generation changed insufficiently. For example, 5 min of lag-time from the second to seventh minute in comparison to the period from the 52nd to 57th minute is supposed to imply the scavenging of the same number of free radicals; in this case, lag-time duration is a reliable parameter. Of course, that was not the case in our assay, as the ABTS^•+^ accumulation rate changed to an extent which cannot be ignored (Figure 2a); thus, we decided to modify this parameter.

We therefore transformed the lag-time duration into the ABTS^•+^ concentration, which was expected to accumulate by the end of every certain lag-time, giving the “Scavenged ABTS^•+^ mixture effect”. Undoubtedly, this method has shortcomings; most importantly, the rate of ABTS^•+^ generation can technically speaking be influenced by antioxidants, and we assumed that this influence was absent or negligible. Despite this, this approach is much more reliable, although it considerably decreased the calculated synergy for combinations (Table 3). On the other hand, this allowed us to avoid the overestimation of mixture effects; thus, we can state with certainty that combinations for which we observed considerable synergy are synergistic, irrespective of the calculation or simulation approaches.

### 4.2. Evaluation of the Antioxidant Capacity of Individual Compounds

Both in the decolorization assay and lag-time assay, the antioxidant capacity of individual compounds demonstrated the expected general tendency, with glutathione and quercetin revealing the lowest and the highest scavenging activity, respectively. However, the absolute amount of ABTS^•+^ molecules captured by one molecule of antioxidant was different. In the decolorization assay, this was excessive both in the case of flavonoids or glutathione and could be ascribed to the occurrence of various sideline reactions. Flavonoids demonstrated high *n*-values even after 1 min of incubation (for example, almost nine molecules of ABTS^•+^ per molecule of quercetin). The glutathione *n*-value after 1 min incubation—2.3 ABTS^•+^ molecules—was also far from the physiologically plausible stoichiometric factor 1 and reached 3.5 ABTS^•+^ molecules by the 30th min. The latter might be explained by the oxidation of glutathione not only to glutathione disulfide but further to sulfinic/sulfenic/sulfonic acids or to the formation of adducts with ABTS^•+^ and was previously observed by several authors [55,80,81,82,83,84]. Moreover, it was recently demonstrated that ABTS^•+^ bleaching by glutathione disulfide substantially evolves at the elevated temperatures [84].

For the lag-time assay, the *n*-values appeared to be much more adequate. Presumably, the antioxidant/ABTS^•+^ ratio plays a critical role here. In the decolorization assay, the ABTS^•+^ is in great excess, leading to high stoichiometries and stimulating all the possible reactions with ABTS^•+^. In the lag-time assay, the opposite situation—i.e., an excess of antioxidant during lag-time and gradual inflow of ABTS^•+^—results in more plausible stoichiometries for most of the examined antioxidants. The recently published review on the reaction pathways involved in the ABTS/PP assay has shed some light on the possible reasons for the high stoichiometries [58]. Apparently, phenolics are not only oxidized when treated by ABTS^•+^ but also form coupling adducts, which can undergo further oxidative degradation, with the formation of hydrazinediylidene-like and imine-like adducts [85,86,87,88,89,90,91,92,93,94]. The complex biphasic pattern reported previously [72,80,95,96,97] suggests the fact that oxidation intermediates of antioxidants subsequently capture ABTS^•+^ radical–cations, thus elevating the total antioxidant capacity; moreover, their contribution can be rather substantial [78].

### 4.3. Evaluation of the Antioxidant Capacity of Flavonoid–Glutathione Combinations

The chemistry of antioxidant effect of a certain antioxidant is presumably obvious: An antioxidant reacts with a free radical thus reducing it, and in its turn is supposed to get converted into a low-reactive oxidized form (neutral or radical). Radical scavenging is a high rate process; regarding, the absolute amounts of reacting species, this process could be observed to be similar to, for example, an acid-base interaction. Obviously, no matter whether sodium hydroxide alone reacts with hydrochloric acid or in a mixture with another alkali, if the acid is in excess. sodium hydroxide neutralizes the same amount of acid, and the stoichiometry does not change. This situation is generally reproduced by various total antioxidant capacity assays, where the total effect of a mixture can be predicted by summing effects of individual compounds in it. The total stoichiometry of a reaction for an antioxidant might not be influenced by a second antioxidant apart from the case when the antioxidants or their oxidized forms interact between themselves. This might lead to different transformations: If the low-reacting sites are converted to high-reacting ones, or a stronger antioxidant is regenerated by the so-called “sacrificial” antioxidant, the synergistic action is observed, and vice versa for subadditivity. 

We calculated synergistic or subadditive effects for combinations by simply comparing their effects with the sum of the antioxidant capacities of their components, as previously done using different antioxidant capacity assays [98,99,100,101,102,103,104,105,106]. In the decolorization assay, the antioxidant capacity of individual compounds generally demonstrated a linear dependence on concentration. The interaction stoichiometry varied insufficiently at different concentrations of both radical or antioxidant: This was demonstrated on the example of quercetin at different concentrations, or the same quercetin concentration with different initial ABTS^•+^ radical–cations concentrations. Furthermore, in the lag-time assay, there are no sensible limits for lag-time duration; thus, we avoided using the sophisticated calculations sometimes applied for mixture effect estimations, which have been mostly focused on drugs or enzyme-related compounds, such as Webb’s method or similar [57,100,107,108,109,110]. 

Actually, the results for all the tested combinations in the decolorization assay were discouraging. Of course, we could speculate about the tendencies and subtle differences between flavonoid combinations and the various ratios of combinations observed (Table 2), but they all were in such a narrow range that the mean value for all the tested combinations’ subadditive effect, at −9%, had a standard deviation of only ±3%. Similar to our results, additive or usually light subadditive antioxidant effects were observed when applying end-point ABTS^•+^-based decolorization assays; e.g., for different combinations of various phenolics [102]; a wide set of different antioxidants, such as ascorbic acid, glutathione, quercetin, and uric acid [81]; human blood plasma combined with flavonoids, incl. quercetin and rutin [111]; binary and ternary mixtures of gallic, ferulic and caffeic acids [112]; and flavonoids and ascorbic acid [113]. Two possible reasons for this subadditivity manifestation were proposed: The formation of non-reactive adducts and regeneration of less active antioxidants by more active antioxidants [81]. In addition, we hypothesized that a sufficient excess of ABTS^•+^ in the decolorization assay incubation medium could prevent most of the possible interactions between components, as the short-living flavoxyl or glutathionyl radicals were much more likely to meet ABTS^•+^ radicals than each other. 

This prompted us to examine the cause of the inconsiderable subadditivity, and we suggest that the cause was the methodology design. To check this hypothesis, the lag-time assay was applied for the same combinations at different ratios. 

In contrast to the decolorization assay, in which we failed to reveal sufficient mixture effects, here we could observe considerable synergy and different tendencies of dependence of the mixture effect on the component ratios (Table 3). 

As long as the structurally related flavonoids were considered, namely two flavonols quercetin and morin with different types of hydroxylation of the B ring (catechol-like and resorcinol-like, respectively)—quercetin glycoside rutin, and flavanonol taxifolin—we performed their comparative analysis.

Several tendencies can be regarded as they have different structural fragments in the B and C-rings (Figure 1). The catechol-like hydroxylation of the B-ring is possibly important for the manifestation of synergy in combinations with glutathione. Only in the case of morin, which has a resorcinol-like hydroxylation of the B-ring, was no synergy observed. Taxifolin, whose specific structural feature is the saturated character of the C2–C3 bond and alcohol-like hydroxyl at C3 in the C-ring, demonstrated the most considerable synergy. A little less synergy was found for rutin–glutathione and quercetin–glutathione combinations, at up to 71% and 51%, respectively. Both have a C2–C3 double bond and phenol-like hydroxyl at C3, although this is glycosylated in the case of rutin. The ratio-dependence of the mixture effect was shown for combinations of taxifolin or quercetin with glutathione: The combinations with a moderate prevalence of glutathione content, from 1:8 to 1:16, appeared to be more synergistic (up to 112%). The most pronounced synergy effect in taxifolin–glutathione may be a consequence of the fact that taxifolin has the lowest redox potential and is closest to that of the reversible redox system GSH–GSSG [19,114,115]. 

### 4.4. Mixture Effect Manifestation and Calculation Approaches

This is not the first work on the assessment of the antioxidant activity of binary flavonoid–glutathione. Pereira et al. [108] studied mixture effects of various flavonoids with glutathione in another discoloration assay, based on the 2,2-diphenyl-1-picrylhydrazyl radical (DPPH^•^). A possible chemical basis of the subadditive, additive and synergistic effects was suggested and comprehensively discussed as well as probable products of flavonoid–glutathione interaction, which were mainly ascribed to the formation of glutathione adducts with flavonoid quinones, which was discovered previously [116,117,118,119,120,121,122,123,124,125,126]. The authors deduced that the presence of a catechol group in the B ring is essential for synergisms with GSH; this conclusion was partly confirmed in our study as well. However, there were certain methodological issues; the most important was the sufficient non-linearity of the DPPH^•^ scavenging capacity from antioxidant concentrations observed for both the flavonoids and GSH. This phenomenon was then demonstrated to be common for various other antioxidants in DPPH^•^ assays [127,128]. To overcome its impact on the mixture effect, Webb’s simulation was carefully adjusted for all the results: effects from subadditive to synergistic (up to 34%) were calculated. Nonetheless, some uncertainty still persists, as any simulation of this kind might lead to overestimation. The lack of unity in the mixture effect calculation approach is confirmed by the fact that several other researchers who have applied the DPPH^•^ assay to elucidate the effect of combination directly compared the antioxidant capacity of a combination with the theoretical sum of the contribution of its antioxidants, which in turn could lead to its underestimation [129,130,131]. Here, we tried to avoid the hyperbolic character of our dose–response curves for flavonoids and glutathione by extending the incubation time (up to 30 min) as we were aware of the recently reported possible dependence of antioxidant quenching stoichiometry on its concentration [69,72]. Although we can claim that we succeeded, almost all of our curves might be described by both linear or power functions with very close correlation coefficients (Appendix A). Possibly, we could therefore also apply Webb’s simulation for our decolorization assay results, thereafter shifting them towards synergistic effects. Such calculation-dependent alterations of combination effects mean that any statements about synergy or subadditivity need to be confirmed by applying different methods and be widely discussed. That is why as to our opinion, the introduction of any additional complexities such as simulations and the like is better avoided. Likewise, we tended to base all our conclusions regarding the decolorization assay results on traditional calculation approaches. In our case, the lag-time approach was adjusted for combinations as alternative and much more explicit effects (up to 83% or 111% synergy, if different calculation strategies were used) were demonstrated, even though a very rigorous mixture effect calculation approach was applied.

## 5. Conclusions

Here, the influence of the modeled system and apparently favorable conditions for the unhindered manifestation of combination effects were carefully demonstrated. Based on the example of ABTS-based assays, we showed that not only the model radical but also calculation strategies and—most of all—the circumstances under which the radical comes into the incubation medium influence the mixture effect. Namely, the same system consisting of an ABTS^•+^ radical–cation and antioxidant combinations was applied in two different ways, each giving totally different results: a great excess of ABTS^•+^ from the very beginning of incubation in the end-point decolorization assay, and the steady-inflow of ABTS^•+^ to the mixture of antioxidants in the lag-time assay. A considerable ratio-dependent synergistic effect was demonstrated for flavonoids with glutathione, which was up to 112% for taxifolin–glutathione combinations. In summary, the gradual ABTS^•+^ inflow allowed the components to interact, thus resulting in high synergies, instead of each component experiencing deeper oxidation separately from each other. It seems that the lag-time strategy can be much more promising in combination effects studies. In contrast, the end-point decolorization assay appeared to be less sensitive to manifestations of mixture effects, therefore suiting the estimation of the total antioxidant capacity of complex combinations. 

## Figures and Tables

**Figure 1 antioxidants-09-00695-f001:**
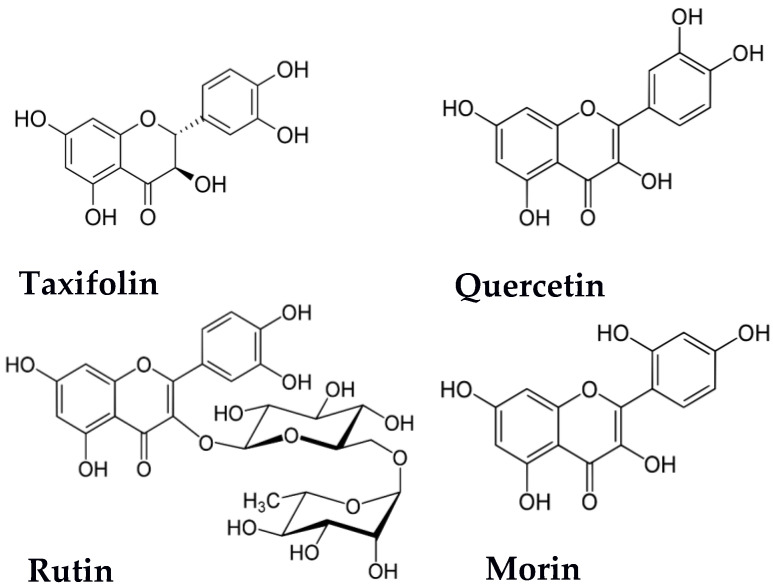
Structures of flavonoids; components of the studied combinations.

**Figure 2 antioxidants-09-00695-f002:**
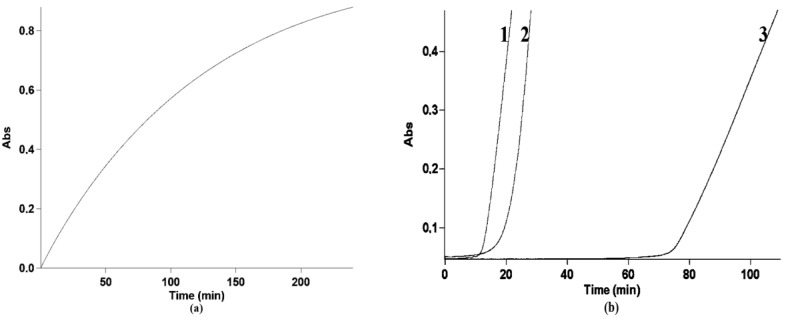
The averaged kinetic curves of the 2,2′-azino-bis(3-ethylbenzothiazoline-6-sulfonic acid radical (ABTS•+) accumulation: (**a**) Without antioxidants or (**b**) in the presence of taxifolin (1) and glutathione (2), and their combination at a ratio of 1:8 (3). (**a**) Averaged from seven runs at the initial concentrations of ABTS and PP of 1.21 mM and 0.43 mM, respectively; λ 730 nm, pathlength 0.1 cm. (**b**) Taxifolin 20 µM (1), glutathione 160 µM (2), and their mixture at 20 + 160 µM (3), respectively. λ 730 nm, pathlength 1 cm.

**Table 1 antioxidants-09-00695-t001:** The *n*-values for flavonoids and glutathione at different incubation times. All *n*-values were statistically different (*p* < 0.05).

Component	*n*-Value ± SD (% of the Total *n*-Value) ^1^			
	1 min	10 min	20 min	30 min
Taxifolin	4.0 ± 0.4 (59.8 ± 4.3%)	6.1 ± 0.5 (93.6 ± 3.1%)	6.5 ± 0.6 (98.5 ± 1.9%)	6.8 ± 0.6
Quercetin	8.9 ± 1.0 (71.9 ± 5.5%)	11.1 ± 0.9 (93.7 ± 1.4%)	11.9 ± 1.0 (98.5 ± 0.7%)	12.4 ± 1.0
Rutin	5.7 ± 0.7 (60.9 ± 5.1%)	8.5 ± 0.7 (93.9 ± 1.6%)	9.0 ± 0.7 (98.8 ± 1.0%)	9.3 ± 0.6
Morin	7.9 ± 1.1 (68.9 ± 4.8%)	10.1 ± 1.1 (92.2 ± 2.1%)	11.0 ± 1.2 (98 ± 1.0%)	11.5 ± 1.3
Glutathione	2.3 ± 0.2 (65.0 ± 4.2%)	3.1 ± 0.3 (92.6 ± 2.6%)	3.4 ± 0.3 (98.6 ± 1.5%)	3.5 ± 0.3

^1^ The *n*-value after 30 min incubation was taken as 100%.

**Table 2 antioxidants-09-00695-t002:** Antioxidant mixture effects of the flavonoid–glutathione combinations in the decolorization assay.

Combination Ratio	Traditionally Calculated Mixture Effect, %				Webb’s Simulation Mixture Effect, %			
Taxifolin–glutathione								
1:1.1 (3.8 µM + 4.2 µM)	−11.15	±	2.76	b	2.92	±	5.91	a
1:5.2 (1.8 µM + 9.4 µM)	−9.92	±	3.88	a	2.90	±	3.67	a
1:9.9 (1.2 µM + 11.9 µM)	−8.74	±	1.18	b	1.45	±	1.60	a
1:15.9 (0.7 µM + 11.1 µM)	−8.45	±	2.17	b	−2.21	±	2.41	a
Quercetin–glutathione								
1:1.1 (2.3 µM + 2.6 µM)	−2.90	±	3.44	a	10.62	±	5.23	b
1:5.0 (1.3 µM + 6.5 µM)	−7.15	±	3.20	d	4.81	±	0.29	a
1:9.6 (0.8 µM + 7.7 µM)	−5.64	±	4.14	a	3.06	±	6.06	a
1:14.3 (0.7 µM + 10 µM)	−10.51	±	1.73	a	−2.28	±	2.68	a
Rutin–glutathione								
1:1.0 (2.2 µM + 2.3 µM)	−12.92	±	1.67	d	−3.75	±	2.74	a
1:5.2 (1.4 µM + 7.3 µM)	−8.36	±	2.44	a	4.92	±	1.07	a
1:10.4 (0.9 µM + 9.4 µM)	−9.50	±	3.84	b	0.65	±	4.59	a
1:15.9 (0.7 µM + 11.1 µM)	−7.98	±	1.17	b	0.77	±	1.05	a
Morin–glutathione								
1:1.0 (3.4 µM + 3.4 µM)	−10.13	±	1.03	a	3.89	±	1.70	a
1:5.1 (1.5 µM + 7.7 µM)	−12.90	±	2.94	a	2.37	±	1.51	a
1:9.6 (1.1 µM + 10.6 µM)	−14.49	±	1.60	a	−0.61	±	3.29	a
1:16.1 (0.7 µM + 11.3 µM)	−10.87	±	3.54	a	0.32	±	2.54	a

a—statistically insignificant, *p* > 0.05; b, d—statistically significant with *p*-values less than 0.05 and 0.001, respectively.

**Table 3 antioxidants-09-00695-t003:** Mixture effects of flavonoid–glutathione combinations in the lag-time assay.

Ratio	Lag-Time Mixture Effect, %				Scavenged ABTS^•+^ Mixture Effect, %			
Taxifolin–glutathione								
1:1 (40 µM + 40 µM)	21.47	±	7.05	b	12.85	±	5.3	a
1:4 (10 µM + 40 µM)	52.35	±	6.19	c	43.61	±	5.41	c
1:8 (10 µM + 80 µM)	101.67	±	4.47	d	81.89	±	3.39	d
1:12 (10 µM + 120 µM)	108.19	±	6.46	c	83.23	±	5.63	c
1:16 (10 µM + 160 µM)	111.57	±	10.24	c	81.96	±	7.52	c
1:20 (10 µM + 200 µM)	94.49	±	3.1	d	66.86	±	2.14	d
1:30 (10 µM + 300 µM)	84.93	±	9.21	c	52.48	±	5.81	c
1:50 (10 µM + 500 µM)	69.55	±	5.84	c	34.49	±	2.37	c
Quercetin–glutathione								
1:4 (10 µM + 40 µM)	30.72	±	14.44	a	21.94	±	12.08	a
1:8 (5 µM + 40 µM)	34.71	±	10.97	b	26.62	±	9.92	b
1:12 (5 µM + 60 µM)	44.01	±	8.56	b	32.99	±	7.14	b
1:16 (5 µM + 80 µM)	51.28	±	9.21	b	38.38	±	8.55	b
1:20 (5 µM + 100 µM)	40.40	±	7.92	b	28.22	±	5.58	b
Rutin–glutathione								
1:2.5 (16 µM + 40 µM)	62.55	±	16.04	b	48.45	±	13.51	b
1:5 (8 µM + 40 µM)	70.84	±	17.57	b	58.11	±	14.94	b
1:7.5 (8 µM + 60 µM)	65.80	±	3.23	d	52.17	±	2.33	d
1:10 (8 µM + 80 µM)	70.36	±	8.39	c	54.48	±	6.83	c
1:12.5 (8 µM + 100 µM)	67.76	±	8.46	c	51.26	±	6.83	c
Morin–glutathione								
1:4 (10 µM + 40 µM)	−3.30	±	1.13	b	−6.84	±	0.98	c
1:8 (5 µM + 40 µM)	−4.51	±	8.25	a	−7.34	±	7.7	a
1:12 (5 µM + 60 µM)	−3.50	±	1.73	a	−6.46	±	1.77	b
1:16 (5 µM + 80 µM)	−3.60	±	2.24	a	−6.56	±	2.19	b
1:20 (5 µM + 100 µM)	−6.52	±	3.04	a	−7.32	±	3.78	a

a—statistically insignificant, *p* > 0.05; b, c, d—statistically significant with *p*-values less than 0.05, 0.01 and 0.001, respectively.

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
