# Peer review of "Flavonoids with Glutathione Antioxidant Synergy: Influence of Free Radicals Inflow"

_antioxidants, 2020, doi:10.3390/antiox9080695_

Round 1

Reviewer 1 Report

In this manuscript, the authors used two different versions of TEAC method to assess the antioxidant synergy of glutathione-phenols mixtures.

In my opinion, the paper has to be improved in some points.

First, it is not clear if the lag-time method has been previously validated and, in the case, what this study adds to the previous studies in terms of method optimization.

Then, it is not explain the reason/advantageous to use Webb’s simulation versus traditional calculation. Indeed, in the bibliographic background no information about Webb calculation is reported.

I also suggest modifying the data presentation for making the results more affordable to the reader. For example, some results, such as those relative to ABTS bleaching, the influence of ABTS concentration and antioxidant addition order, could be moved in a supplementary section.

Author Response

Dear reviewer,

Thank you for your assessment of our work, we believe that reviewing and revision process will help to make our manuscript considerably better.

Regarding your Comments and Suggestions please see below our answers and considerations (in green):

  1. “In this manuscript, the authors used two different versions of TEAC method to assess the antioxidant synergy of glutathione-phenols mixtures.

In my opinion, the paper has to be improved in some points.

First, it is not clear if the lag-time method has been previously validated and, in the case, what this study adds to the previous studies in terms of method optimization.”

The lag-time method was previously validated, we refer to the work where it was first presented, please see:

  1. Ilyasov, I.R., Beloborodov, V.L. & Selivanova, I.A. Three ABTS•+radical cation-based approaches for the evaluation of antioxidant activity: fast- and slow-reacting antioxidant behavior. Chem. Pap.72, 1917–1925 (2018). https://doi.org/10.1007/s11696-018-0415-9

In this work, we utilized this approach without any sufficient changes and therefore did not add to the previous studies in terms of method optimization.

  1. Then, it is not explain the reason/advantageous to use Webb’s simulation versus traditional calculation. Indeed, in the bibliographic background no information about Webb calculation is reported.

Generally, in our careful opinion, there is no advantage to use Webb’s simulation (apart from the fact that in this case the results are shifted towards synergy – of course, if it can be accounted as an advantage). We strongly prefer traditional calculation and all our main conclusions were made on their basis, however, as long as some other works are based on the use of this type of simulation and it is discussed in some relevant reviews we could not avoid discussing it in our manuscript. Below are some of them:

  1. Webb JL. Enzyme and metabolic inhibitors. Vol.1. General principles of inhibition. New York: Academic Press, 1963. 55–79 p.
  2. Olszowy-Tomczyk M. Synergistic, antagonistic and additive antioxidant effects in the binary mixtures. Phytochemistry Reviews. Springer; 2020. p. 1–41.
  3. Chou T-C, Talalay P. Quantitative analysis of dose-effect relationships: the combined effects of multiple drugs or enzyme inhibitors. Adv Enzyme Regul. 1984 Jan;22(C):27–55.
  4. Pereira R, Sousa C, Costa A, Andrade P, Valentão P. Glutathione and the Antioxidant Potential of Binary Mixtures with Flavonoids: Synergisms and Antagonisms. Molecules. 2013 Jul;18(8):8858–72.

Anyway, we say that the simulation use can lead to overestimation: line 471 “Nonetheless, some uncertainty still persists as any simulation of this kind might lead to overestimation“.

In addition, we added the following phrase (see line 484) “That is why as to our opinion, the introduction of any additional complexities such as simulations and the like is better avoided. Likewise, we tended to base all our conclusions regarding the decolorization assay results on traditional calculation approaches.”

To sum up, our point of view is that the use of the traditional calculation approach is to be preferred.

  1. I also suggest modifying the data presentation for making the results more affordable to the reader. For example, some results, such as those relative to ABTS bleaching, the influence of ABTS concentration and antioxidant addition order, could be moved in a supplementary section.

We agree that possibly our manuscript was overloaded with extensive experimental data, so we decided to move tables 1-3 and all figures except for fig.1 and fig.6 to supplementary section.

Reviewer 2 Report

The subject of antioxidants is polyhedral, complex and difficult to deal with, both when it comes to simple compounds or binary mixtures or "complex network" given the possible cooperative interactions of their components, in the latter cases. The authors address in this paper the study of the behaviour of binary mixtures of flavonoids with glutathione. It seems to us that the counterpart of flavonoid is a good choice, as it form part of the main redox cell buffer. The paper carried out involves a systematic and detailed study of the synergistic or subadditive effects of these mixtures in which both the structure of the flavonoids and the molar ratio of the components seem to exert a key action. It is an interesting academic exercise far from the real world, but the results obtained can be very useful to other groups of researchers working on the subject. The main objection we raise (and perhaps the only one) is that the paper is very extensive, paying a lot or too much attention to detail in its writing, and perhaps its length could be considerably reduced without losing its scientific content. In particular, maintaining the literature, most of the figures and graphs could be omitted (except perhaps Fig. 1 and 6), I think, and presented as supplementary material.

Some minor order issues are listed below:

Line 90: I do not like the use of dimethylformamide as a minor solvent in stages that ultimately claim to have an in vivo applicability, even if it does not interfere with the assays. Perhaps there is no choice but to use it given the poor solubility of rutin and the magical solvent properties of dimethylformamide.

Line 111: The sentence of "N-value and mixture effect calculations" is isolated from the rest. It may be in bold print.

Line 372: Note "dissolute" at the end of the line. Maybe perhaps "dissolved"?

In short, the authors have made a great effort worthy of praise. It is, in my opinion, a detailed study, very well documented, that deserves publication in ANTIOXIDANTS, if the suggestions indicated above are followed.

Author Response

Dear reviewer,

Thank you for your assessment of our work, we believe that reviewing and revision process will help to make our manuscript considerably better.

Regarding your Comments and Suggestions please see below our answers and considerations (in green):

  1. The subject of antioxidants is polyhedral, complex and difficult to deal with, both when it comes to simple compounds or binary mixtures or "complex network" given the possible cooperative interactions of their components, in the latter cases. The authors address in this paper the study of the behaviour of binary mixtures of flavonoids with glutathione. It seems to us that the counterpart of flavonoid is a good choice, as it form part of the main redox cell buffer. The paper carried out involves a systematic and detailed study of the synergistic or subadditive effects of these mixtures in which both the structure of the flavonoids and the molar ratio of the components seem to exert a key action. It is an interesting academic exercise far from the real world, but the results obtained can be very useful to other groups of researchers working on the subject. The main objection we raise (and perhaps the only one) is that the paper is very extensive, paying a lot or too much attention to detail in its writing, and perhaps its length could be considerably reduced without losing its scientific content. In particular, maintaining the literature, most of the figures and graphs could be omitted (except perhaps Fig. 1 and 6), I think, and presented as supplementary material.

Thank you for your high assessment of our work.

We agree that possibly our manuscript was overloaded with extensive experimental data, so we decided to move tables 1-3 and all figures except for fig.1 and fig.6 to supplementary section.

  1. Some minor order issues are listed below:

Line 90: I do not like the use of dimethylformamide as a minor solvent in stages that ultimately claim to have an in vivo applicability, even if it does not interfere with the assays. Perhaps there is no choice but to use it given the poor solubility of rutin and the magical solvent properties of dimethylformamide.

Indeed, we either prefer water or alcohols to dissolve flavonoids, but it was difficult for rutin even if heating. That is why only we took a small amount of N,N-dimethylformamide to initially dissolve rutin and then added a ten-fold amount of ethanol as the main solvent (dimethylformamide:ethanol ratio was 1:10). So the final amount of N,N-dimethylformamide used was rather small.

Line 111: The sentence of "N-value and mixture effect calculations" is isolated from the rest. It may be in bold print.

It was our mistake, of course, it was supposed to be in bold (as well as it is in line 147 for lag-time assay). We corrected it (see line 115). Thank you for your attention.

Line 372: Note "dissolute" at the end of the line. Maybe perhaps "dissolved"?

Thank you again, of course, “dissolved” is to be used. We corrected it (see line 316).

Reviewer 3 Report

The study addressed the antioxidant interaction of the flavonoid-glutathione composition with different ratios using two different 2,2’-azino-bis(3-ethylbenzothiazoline-6-sulfonic acid radicals cations based ways. The scientific quality of the study is very adequate, however, the English language should be revised

line 65: These flavonoids are textbook examples of natural antioxidants…Please remove textbook and rephrase the whole sentence.

Author Response

Dear reviewer,

Thank you for your assessment of our work, we believe that reviewing and revision process will help to make our manuscript considerably better.

Regarding your Comments and Suggestions please see below our answers and considerations (in green):

  1. The study addressed the antioxidant interaction of the flavonoid-glutathione composition with different ratios using two different 2,2’-azino-bis(3-ethylbenzothiazoline-6-sulfonic acid radicals cations based ways. The scientific quality of the study is very adequate, however, the English language should be revised

Thank you for your high assessment of our work. We applied MDPI English editing service to check grammar, spelling, punctuation and make some improvement of style where necessary. Hope this improved our manuscript's English language. Please see the certification enclosed.

  1. line 65: These flavonoids are textbook examples of natural antioxidants…Please remove textbook and rephrase the whole sentence.

We tried to carefully rephrase this sentence together with English editing service as follows (see line 66):

“These flavonoids are well-known natural antioxidants whose potential health benefits have attracted a great deal of interest in their studies as prospective drugs”

Round 2

Reviewer 1 Report

The authors revised the manuscript according to the reviewer's comments. I only suggest to mention some literature information about the use of Webb's simulation in the introduction.